# Slump in Hospital Admissions for Stroke, a Fact of an Uncertain Nature That Requires Explanation

**DOI:** 10.3390/brainsci11010092

**Published:** 2021-01-13

**Authors:** José M. Ramírez-Moreno, Juan Carlos Portilla-Cuenca, Roshan Hariramani-Ramchandani, Belen Rebollo, Inés Bermejo Casado, Pablo Macías-Sedas, David Ceberino, Ana M. Roa-Montero, Alberto González-Plata, Ignacio Casado, Luis Fernández de Alarcón

**Affiliations:** 1Department of Neurology, University Hospital of Badajoz, 06080 Badajoz, Spain; roshan.hariramani@salud-juntaex.es (R.H.-R.); belen.rebollo.93@gmail.com (B.R.); pmsedas@gmail.com (P.M.-S.); david_cebe@hotmail.com (D.C.); anaroam82@hotmail.com (A.M.R.-M.); gplata.alberto@gmail.com (A.G.-P.); 2Department of Biomedical Sciences, Extremadura University, 06080 Badajoz, Spain; 3Multidisciplinary Research Group of Extremadura (GRIMEX), 06700 Villanueva de la Serena, Spain; 4University Institute of Biosanitary Research of Extremadura (INUBE), 06080 Badajoz, Spain; ignacio.casado@salud-juntaex.es; 5Department of Neurology, University Hospital of Cáceres, 10003 Cáceres, Spain; portilla7819@yahoo.es (J.C.P.-C.); icasadon@gmail.com (I.B.C.); 6Department of Radiology, University Hospital of Badajoz, 06080 Badajoz, Spain; lfdalarcon@gmail.com

**Keywords:** stroke, pandemic, reperfusion, COVID-19, public health, stroke unit

## Abstract

(1) Background: The impact of the health crisis caused by coronavirus disease 2019 (COVID-19) has provoked collateral effects in the attention to pathologies with time-dependent treatments such as strokes. We compare the healthcare activity of two stroke units in the same periods of 2019 and 2020, with an emphasis on what happened during the state of alarm (SA). (2) Materials and methods. Hospitals in the region implemented contingency plans to contain the pandemic; in this planning, the stroke units were not limited in their operational capacity. The SA was declared on 15 March and remained in place for 10 weeks. For the analysis, the data were grouped by consecutive calendar weeks. (3) Results. When the SA was declared the number of calls to the emergency telephone went from 1225 to 3908 calls per week (318% increase). However, the activation of the stroke code went from 6.6 to 5.0 (*p* = 0.04) and the activity in both stroke units decreased. The largest drop in hospitalizations was for transient ischemic attacks (TIAs) with 35.7% less, 28 vs. 18, (*p* = 0.05). Reperfusion therapies fell by 37.5%; Poisson regression model 0.64; (95% confidence interval (CI), 0.43–0.95). The overall activity of the telestroke suffered a reduction of 28.9%. We also observed an increase in hospital mortality. (4) Conclusion. The excessive duration of the pandemic precludes any hope of resolving this public health crisis in the short or medium term. Further studies should be conducted to better understand the multifactorial nature of this dramatic decline in stroke admissions and its negative impact.

## 1. Introduction

In December 2019, the first identified cases of a new coronavirus, severe acute respiratory syndrome coronavirus 2 (SARS-CoV-2), in Wuhan, China, was followed a pandemic with cases reported in more than 200 countries [1]. The crisis is so severe that it could overwhelm the health care system in many countries [2], and depending on the local intensity of the epidemic, there is a high risk of not applying well-established therapies to patients with prevalent conditions such as cancer, acute myocardial infarction, or acute ischemic stroke [3,4,5].

The 2019 coronavirus disease (COVID-19) pandemic challenges health leaders to make critical and urgent decisions about clinical activities and resource allocation [6]. Its impact on acute stroke care may not be dependent only on health policy decisions made by authorities, but also on the social behavior of the population. How efficient the overall system of care is in providing optimized prehospital triage and equitable access to acute treatment in hospitals will undoubtedly be key [5,7].

This study, which compares the care activity of two consolidated stroke units in the same periods of 2019 and 2020, with emphasis on the state of alarm decreed in our country, pays special attention to the number of activations of the stroke code, admissions for acute cerebrovascular disease, consultations for tele-stroke and the number of reperfusion therapies performed on eligible patients, and includes prognostic variables such as in-hospital mortality and functional status at discharge.

## 2. Materials and Methods

### 2.1. Health System and Stroke Organization

In Extremadura (total population 1.1 million inhabitants), acute stroke care is provided through a network consisting of a stroke center, a stroke unit and 9 hospitals connected by telestroke. The Emergency Medical System is the main activator of the stroke code and the transport of patients guarantees the urgent and prioritized transfer of a patient with suspected stroke to the nearest hospital with adequate diagnostic and therapeutic capacity.

The Hospital Universitario de Badajoz (HUB) is a public university hospital with 915 hospital beds, including 30 intensive-care beds, which serves as a community hospital for a population of 271,885 inhabitants and provides tertiary stroke care to a population of 676,376 inhabitants. Its stroke center has a 6-bed semi-intensive care unit and a multidisciplinary team led by a vascular neurologist. During the year 2019, 476 strokes were attended, and 178 acute phase reperfusion treatments were performed, of which 96 were mechanical thrombectomies.

The Hospital Universitario de Cáceres (HUC) is a public university hospital with 520 hospital beds, including 16 intensive care beds, serving as a community hospital for a population of 195,074 inhabitants and providing tertiary care for acute stroke to a population of 396,487 inhabitants. It has a unit with 4 semi-intensive care beds and a multidisciplinary team led by a vascular neurologist. During the year 2019, 351 strokes were treated, and 84 acute phase reperfusion treatments were performed. 

### 2.2. Planning for Pandemic of Coronavirus Disease 2019 (COVID-19)

In February 2020, the first confirmed cases of patients with COVID-19 were reported. Thereafter, hospitals in the region implemented contingency plans to contain the pandemic, including suspension of non-essential visits and adjustment of clinical services for outpatients and inpatients, as well as progressive opening of quarantine rooms according to care needs. 

Within this overall planning, the region’s stroke units were not constrained in their capacity, and inpatient stroke unit staffing was not modified during the height of the pandemic. Multimodality brain imaging, computed tomography angiography, and endovascular treatment were guaranteed to be available 24/7 for eligible patients. There were also no limitations on the use of telestroke, and all 9 hubs were operational. Outpatient visits were not conducted in person, except in special cases and on demand, and most were conducted by telephone contact. 

Admission of new patients to the stroke units was always conditional on their having no symptoms/signs of infection in order to try to keep the units clean of COVID-19. Therefore, patients with stroke and fever or lung symptoms or pathological radiological findings in the chest were admitted to an isolated area of the hospital and then transferred to the stroke unit once infection was ruled out by a negative SARS-CoV-2 polymerase chain reaction (PCR) test. If the patient was infected by the virus, he was treated and monitored in the COVID area of the hospital.

There were no relevant changes in the stroke care protocol, except for reperfusion treatments in the interventional radiology room, where patients were given rapid COVID-19 serological tests prior to thrombectomy. The staff wore full personal protective equipment in all cases. 

The confinement of the population due to the state of alarm was decreed in our region on 15 March, as in the rest of the Spanish state, and was maintained until 10 May [8]. 

### 2.3. Quality Indicators for Stroke Care and Data Collection

The main variables of the study were: stroke code activations, number of stroke admissions (ischemic, hemorrhagic, transient ischemic attacks or minor stroke), number of tele-strokes attended, reperfusion therapies (intravenous or thrombectomy) performed on eligible patients, severity of stroke on admission according to the National Institute of Health Stroke Scale (NIHSS), functional status at discharge (according to the modified Rankin Scale score), and hospital mortality. Activity data were collected retrospectively from the two stroke units in the region between 1 January and 31 May 2019 and 2020.

The information on the total number of COVID-19 cases in Spain and the Autonomous Community of Extremadura has been obtained from the data provided by the Ministry of Health on its official pandemic website (https://cnecovid.isciii.es/covid19/#documentación-y-datos). The results presented in this panel are obtained from the declaration of COVID-19 cases to the National Network of Epidemiological Surveillance (RENAVE) through the Surveillance System in Spain (SIVIES) web computing platform managed by the National Centre of Epidemiology (CNE). This information comes from the case epidemiological survey that each autonomous community completes after identifying a case of COVID-19.

The data on activations of emergencies and emergency in our autonomous community were obtained by consulting the SITREM^®^ (comprehensive system for emergency treatment) of incidents uploaded into the system between 1 January and 31 May of 2019 and 2020, excluding incidents without participation of the health sector (fires, incidents related to public safety, technical information or road assistance).

To assess the impact of the pandemic, the outcome variables were compared in the following periods: (1) January to May 2019 and 2020, (2) the state of alarm in 2020 (confi2020) versus the same period in 2019 (confi2019), and (3) the state of alarm in 2020 (confi2020) versus the period of normality (norm2020). For the analysis, all data were grouped by consecutive calendar weeks.

### 2.4. Ethical Considerations

The study protocol was not submitted to the local ethics committee and meets the state legal requirements in the field of biomedical research, personal data protection and good clinical practice standards, also as stated in the Declaration of Helsinki. Given the global use of the data, individual patient consent was not seen as necessary.

### 2.5. Statistical Analysis

Dichotomous variables are presented as numbers and percentages. Quantitative variables are presented as means and standard deviations (SD). Intrahospital action times are presented as medians and interquartile ranges (IQR). The chi-square test and Fisher’s test were used to compare qualitative variables. To compare quantitative variables that follow a normal distribution, the Student’s *t*-test or analysis of variance (ANOVA) was used. To compare medians between two groups, the Mann–Whitney U test was used for non-parametric samples and for more than two groups the Kruskal–Wallis test was used. 

To compare the total number of patients receiving reperfusion treatment between the study periods we used Poisson regression models adjusted for age and sex. To establish the existence of correlation between the number of patients attended and treated with reperfusion therapies in the COVID-19 period we used the Pearson correlation coefficient. To make the trend graphs we used the univariate autoregressive integrated moving average (ARIMA) time series modeling procedures and the spectral graphs procedure which is used to identify periodic time series.

All statistical analyses were performed with IBM SPSS Statistics V.22, IBM. A *p* < 0.05 indicated a statistically significant difference.

## 3. Results

Spain, with 265,189 confirmed cases of COVID-19 as of 31 May 2020, has one of the highest burdens of this pandemic in the world. In our area of study (Extremadura) on the same date there were 5604 confirmed cases. In response, the Spanish government declared a state of alarm and confinement of the population on 15 March, when the basic reproductive number of the virus (Rt) reached the 2.7 in the whole of the country and 3.6 in Extremadura.

### 3.1. Stroke Code Activations

The number of calls to the “112” health emergency phone increased during the state of alarm period, going from 1225 weekly calls in the same period in 2019 to 3908 weekly calls in 2020, a percentage increase of 318% in our region. Although the stroke code activation increased in 2020 compared to the previous year by 19% (130 vs. 103 activations in total; *p* = 0.012) (Figure 1A), this number of activations during the 10 weeks of strict confinement fell very significantly, and there was not the expected increase in activations for this period compared to the same period in the previous year (Figure 1B). The average number of weekly stroke codes sent to the two stroke units went from 6.6 in the norm2020 period to 5.0 in the confi2020 period, (*p* = 0.044).

### 3.2. Stroke Admission to Stroke Units

In the confinement phase (15 March–10 May), activity in both stroke units decreased compared to 2019, with a 21% reduction in admitted cases, from 205 cases in 2019 to 162 (*p* = 0.011). Ischemic stroke decreased by 20%, 155 vs. 124 (*p* = 0.039); hemorrhagic stroke by 9.1%, 22 cases vs. 20 without significant differences (*p* = 0.791) and transient ischemic attacks (TIAs) and minor stroke by 35.7%, 28 vs. 18, (*p* = 0.05). In the period corresponding to normality (non-confined) there were no significant differences between 2019 and 2020 in the number of total admissions (149 vs. 133; *p* = 0.215), ischemic stroke (111 vs. 106; *p* = 0.655), hemorrhagic stroke (14 vs. 10; *p* = 0.283) and TIA/stroke minor (23 vs. 17; *p* = 0.171).

No significant differences were observed in the mean age of patients on admission in the confi2019 and confi2020 periods (71.2 (standard deviation, SD 3.5) vs. 70.4 (SD 2.7); *p* = 0.585), nor between the norm2020 and confi2020 periods (*p* = 0.621). There were also no differences observed in the number of males admitted in the confi2019 and confi2020 periods (94 vs. 85; *p* = 0.448); but there were significant differences in the number of females admitted in those periods (74 vs. 55; *p* = 0.015). 

### 3.3. Stroke Severity

No differences were observed in the mean scores of NIHSS at admission in the periods confi2019 and confi2020 (6.7 (SD 1.9) vs. 7.1 (SD1.6); *p* = 0.621), nor between the periods norm2020 and confi2020 (6.5 (SD 1.4) vs. 7.1 (SD 1.6); *p* = 0.621).

### 3.4. Reperfusion Therapies

Regarding reperfusion therapies, thrombolytic therapy was reduced by 51.4% (average patients treated per week in the periods confi2019 and confi2020: 3.5 vs. 1.7; *p* = 0.003). Using age and sex-adjusted Poisson regression models, when we compared the periods confi2019 and confi2020 we observed a significant variation (0.58; (95% confidence interval (CI), 0.34–0.98); *p* = 0.041). However, the mean number of weekly thrombectomies performed by the reference center remained stable with 1.3 cases per week in those periods, with no significant variation observed (0.75; (95% CI, 0.40–1.37); *p* = 0.348). Considering any reperfusion therapy offered to patients with ischemic stroke, a reduction of 37.5% was observed in the confi2020 period (*p* = 0.07). The Poisson regression model shows a significant variation (0.64; (95% CI, 0.43–0.95); *p* = 0.027). Figure 2 visually represents the impact of the pandemic according to the reperfusion and activation treatment modalities of the stroke code.

### 3.5. Use of a Telestroke Service

With respect to the patients attended through the nine existing telestroke hubs in the region; in the period from 1 January to 31 May 2019, 50 patients were attended and in the same period of 2020, 48 patients were attended (*p* = 0.728). However, there was a 28.9% reduction between the confi2019 and confi2020 periods (average patients attended per week: 3.8 versus 2.7, *p* = 0.089), and a 22% reduction if we compare the norm2020 and confi2020 periods (average patients attended per week: 2.7 versus 1.8, *p* = 0.201). 

### 3.6. Stroke Prognosis

Hospital mortality in the confi2019 period was 4.4% (CI 95%: 2.3–8.1) while in the confi2020 period it was 7.4% (CI 95%: 4.2–12.5) (*p* = 0.216). Historical hospital mortality over the last 10 years has varied between 2.5% and 5.1%, an increase in mortality of 3 points seems clinically relevant regardless of the level of significance.

The neurological outcome at discharge, as measured by the Rankin scale in the confi2020 period (median 2.5, interquartile range 1.25) was also slightly worse than in the confi2019 period (median 1, interquartile range 1.25) (*p* = 0.174). 

Basic distribution statistics of the main variables are in Table 1. A summary of the main characteristics of the patients admitted to stroke units and quality measurers of acute stroke care provided during the pandemic in relation to the previous year can be seen in Table 2.

During the entire pandemic period, there was a weak negative correlation between the number of subjects affected by Covid-19 in our region and the number of stroke code activations (R2: −0.09), number of patients admitted for TIA/stroke minor (R2: −0.03), number of patients treated with thrombolytic therapy (R2: −0.409), treated with mechanical thrombectomy (R2: −0.455), and total number of reperfusion therapies (R2: −0.525). See Table 3 for details.

Figure 3 is a visual representation that shows the average value predicted by the ARIMA model of the main variables in relation to the period of time studied. It can be seen that during the confinement of the population in the year 2020, the linear effect observed in a year without critical incidents such as 2019 is clearly lost, producing a fall in the number of activations of the stroke code (Figure 3A), the number of hospital admissions for stroke (Figure 3B), and the number of reperfusion treatments (Figure 3C) and an increase in mortality (Figure 3D).

## 4. Discussion

The severity of the COVID-19 pandemic has had a very important impact on the health system, in many cases leading to a situation of collapse [2]. It has been necessary to make very fast, deep and varying changes in hospitals, emergency systems and primary care, which have had side effects in other serious pathologies [9] and in those that are time-dependent [10,11]. In our study, we observed that during the hardest stage of the pandemic (alarm state/strict confinement), there was a reduction in the number of stroke patients admitted to our health system, associated with a sharp decrease in the number of stroke code activations from the emergency coordination center, a marked decrease in reperfusion treatments, and an increase in hospital mortality. This effect has been significantly greater in women. The issue of gender differences in stroke is not new and these findings are not surprising. It is possible that socioeconomic factors including marital status played a role in the outcomes of this study. Because women live longer than men, they may live alone at the time a stroke occurs and may be socially isolated.

With the information published before the arrival of the pandemic in our country, it seemed possible that the opposite would occur, an increase in the number of acute ischemic strokes and, therefore, of reperfusion treatments due to the state of hypercoagulability induced by the virus which could also condition neurovascular complications related to COVID-19 [12,13]. Paradoxically, our analysis shows the opposite and similar data have been published in other countries [14,15] and regions of our country [5,7,16,17].

While there are no more data on the causes that support these results, the most plausible hypotheses would be the negative effect of social isolation of the elderly population, more vulnerable to stroke, the fear of going to the hospital because of the risk of contagion, perhaps a misinterpretation of the repeated slogans of politicians, media and social networks such as “stay at home”. Despite the opening of specific telephone help lines by COVID-19, many times there has been a collapse of call centers to the emergency services and that could have limited access to the health system, this aspect has been communicated also in other regions of the world [18,19].

It is also the case that epidemiologically the patients with a higher risk of infection and mortality from COVID-19 are older, with greater comorbidity and more risk factors such as hypertension, diabetes, smoking or obesity and therefore at greater risk of stroke [20]. Therefore, the disease caused by COVID-19 should be considered as a competitive risk event when evaluating the reduction in the admission of acute cerebrovascular pathology. The admission in a given period is not a repetitive event, so that if an infectious cause in this case advances to another vascular one to cause the admission of the individual, the rate of hospitalization of the advanced cause will necessarily decrease [21].

Another problem may have been that awareness of stroke warning signs and symptoms among our emergency department colleagues who should activate the stroke code or call neurologists may be suboptimal due to excessive replacement of frontline staff or a preferential effort to prioritize pandemic care [22]. This could also have been the case in the emergency departments of the smaller hospitals in the network and would further explain why stroke activity was dramatically reduced [23,24].

Looking at the problem from another point of view, the reduction in the volume of strokes admitted to the hospital could be related to the way we handle stress. Psychosocial and occupational stress is recognized as a risk factor for vascular disease and stroke [25]. With confinement, life was slowed down and this could reduce the negative effects of stress for many subjects with vascular risk factors, there would be less work activity, more dynamic physical exercise and more hours of sleep, which would induce beneficial changes in blood pressure, vascular tone and reactivity and in the endocrine system [26]. To assume that these physiological changes could have been produced in the confined subjects and that they would have a preventive character is a plausible explanation that, nonetheless, seems unlikely and would explain only a very small part of the facts.

The sharpest reduction in admissions was observed in patients with TIA and minor stroke. This may be due mainly to two facts: first, classification strategies were too restrictive by out-of-hospital medical personnel and, before attending patients at home, attempts were made to call patients to ensure that they should be transferred to hospitals, This could prevent some patients from receiving a more complete evaluation in hospitals when the symptoms were minor, and secondly a perception of lesser severity of the stroke by the population against the virus infection, thus avoiding hospital consultation for minor symptoms. One limitation of our study is that we do not have information on those patients who could have been cared for at home or died at home due to a stroke without being cared for. Several stroke cases have likely been treated at home during the pandemic, but we do not have that data, and our records are not aligned with top-tier health care records.

The reduction in the number of admissions for ischemic stroke and, therefore, also of patients potentially eligible for reperfusion treatment could explain the absolute reduction in the number of reperfusion therapies performed during what we call the confi2020 period and which corresponds to the alarm state. Other authors have reported that during the pandemic there have been important delays in the time of attention to stroke, both in the time of arrival at the hospital from the onset of symptoms [16], and in hospital care [18], a fact that we cannot confirm in our study but is our subjective perception. In addition, patients who live alone at home may be less likely to be seen by family and friends early when they have the first symptoms, and when the alarm is activated it is too late to receive therapy and they may even die without assistance [27]. 

Along the same lines as what happened in our region, hospitalizations for stroke or transient ischemic attacks in other countries fell during the worst period of the COVID-19 outbreak [28]. Intravenous thrombolytic treatments also decreased, while endovascular treatments remained unchanged and even increased in the area of maximum expression of the outbreak [29]. The limited hospitalization of less severe patients and delays in hospital admission, due to the overload of the emergency system by COVID-19 patients, may explain these data.

In our study, we observe a higher hospital mortality in that period, and a worse high functional situation, although not statistically significant, seems to us clinically relevant, and an unfortunate indirect consequence of the reactive measures promoted against the pandemic. Increased severity of a stroke on admission, delayed care, and lower rates of reperfusion therapy, may explain the worse outcomes. 

Health authorities and professionals should continue to encourage the population to continue to seek urgent care if they experience symptoms of acute stroke regardless of whether they are transient or permanent [30]. There is always time to improve health education, especially for those at high risk of stroke, by helping them to recognize symptoms and call emergency medical services immediately. A clear message about where to call and where to go depending on the problems related or not related to COVID-19 is critical to diminish the fear of the population, explaining that clean circuits are secured and that they will not be infected when they go to the hospital looking for help for a serious acute illness like stroke [11,18]. Probably, and according to other authors, it is also time to have separate specific telephone help lines for serious illnesses that are time-dependent that would help people to request assistance and avoid delays in treatment [17]. 

## 5. Conclusions

In our region with a consolidated stroke care network, we observed during the COVID-19 pandemic a strong decrease in hospital admissions for stroke, especially when they were milder and affect women. It is not clear why there has been this decline, but it is likely that patient fears of hospitalization and infection, social distancing, isolation, and underreporting of symptom severity are playing a significant role in decreasing outpatient assessments for acute stroke and TIA. The excessive duration of the pandemic precludes any hope of resolving this public health crisis in the short or medium term. Further studies should be conducted to better understand the multifactorial nature of this dramatic international decline in stroke admissions and its negative impact.

## Figures and Tables

**Figure 1 brainsci-11-00092-f001:**
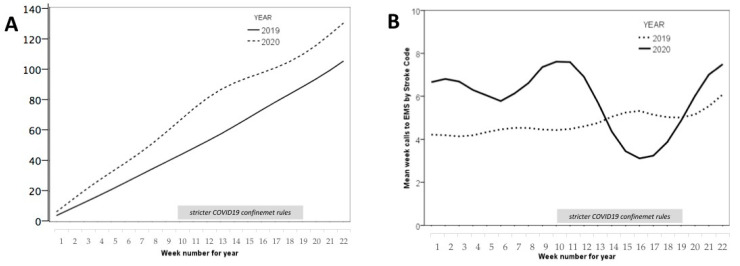
Stroke code activation. (**A**) Cumulative stroke code activations to stroke units according to the year. (**B**) Mean week calls to emergency medical services (EMS) by stroke code according to the year.

**Figure 2 brainsci-11-00092-f002:**
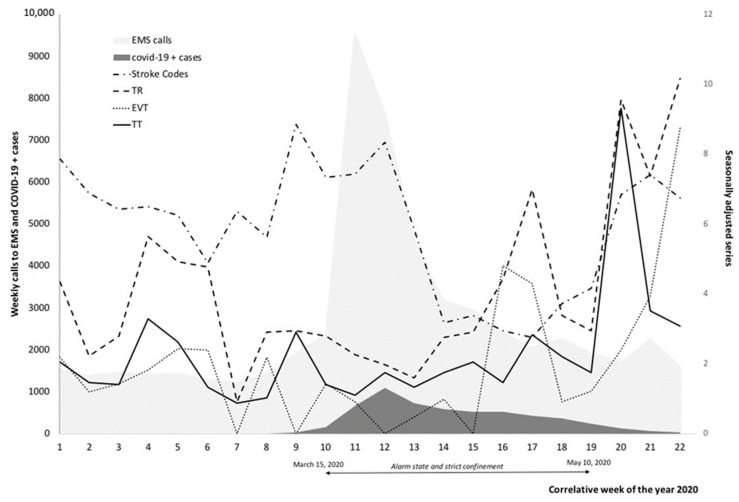
Graphic representation to the impact of the pandemic according to the reperfusion and activation treatment modalities of the stroke code. EMS: Emergency medical services, TT: N° Thrombolytic therapy, EVT: N° Endovascular therapy, RT: N° Reperfusion therapy.

**Figure 3 brainsci-11-00092-f003:**
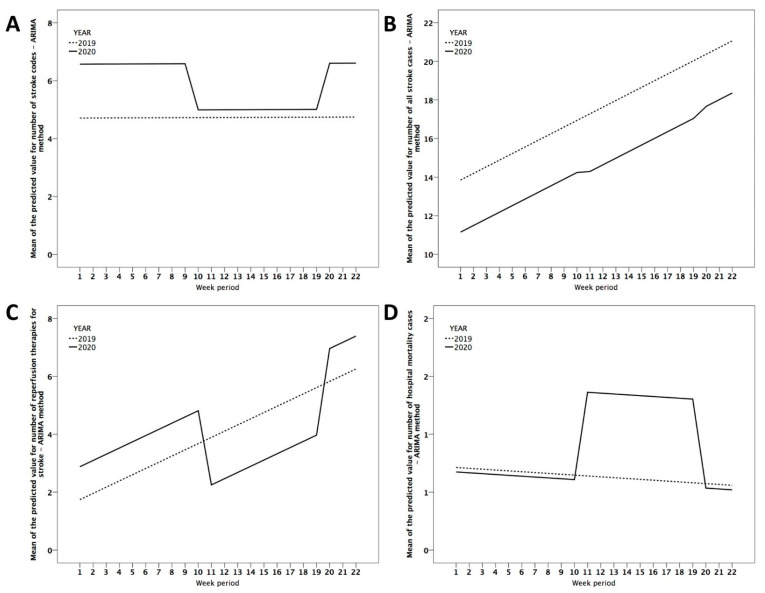
Visual representation that shows the average value predicted by the univariate autoregressive integrated moving average (ARIMA) model of the main variables in relation to the period of time studied. Figures are explained in the text.

**Table 1 brainsci-11-00092-t001:** Basic distribution statistics per week activity in the two stroke units: Variable Name, Maximum value (max), 75th Percentile (q75), 50th Percentile (median), 25th Percentile (q25), Minimum value (min), Mean, Standard Deviation (SD).

Variable	Max	q75	Median	q25	Min	Mean	SD
Age (years old)	77.7	72.9	71.1	68.6	64.8	70.9	2.9
Stroke code activations	9	6	5.1	4.2	3	5.5	1.5
All stroke admissions	28	18	16	13	8	16.1	4.0
Ischemic stroke	23	14	12	9.5	7	12.3	3.3
Hemorrhagic stroke	6	2	1	1	0	1.6	1.3
Transient Ischemic Attack	4	3	2	1	0	1.2	1.2
Tele-stroke consultations	6	3	2	1	0	2.3	1.7
Stroke severity (NIHSS)	11	8	7	5.3	2	6.8	1.8
Thrombolytic therapy	7	3	2	1	0	2.4	2.4
Endovascular therapy	5	2.5	1	1	0	1.5	1.4
Reperfusion therapy	12	5.5	3	2	1	4.0	2.4
In-hospital mortality	3	1	1	0	0	0.8	0.8
Length stay (days)	19	11	9.5	8	6	9.5	2.4

Variable Name, Maximum value (max), 75th Percentile (q75), 50th Percentile (median), 25th Percentile (q25), Minimum value (min), Mean, Standard Deviation (SD), NIHSS: National institute of Health Stroke Scale.

**Table 2 brainsci-11-00092-t002:** Main characteristics of the patients admitted to stroke units and quality measurers of acute stroke care provided during the pandemic in relation to the previous year.

	Jan–May 2019 (Weekly)	Jan–May 2020 (Weekly)	*p*	Confi2019 (Weekly)	Confi2020 (Weekly)	*p*
Calls EMS Center	1245	2631	0.006	1225	3908	0.011
Stroke code activations	4.7	5.9	0.009	4.9	5.0	0.936
All stroke admissions	17.5	14.6	0.020	20.5	16.2	0.011
Ischemic stroke	13.1	11.5	0.097	15.5	12.4	0.039
Hemorrhagic stroke	1.8	1.5	0.441	2.2	2.0	0.791
Transient Ischemic Attack	2.5	1.7	0.026	2.8	1.8	0.086
Tele-stroke consultations	2.4	2.2	0.728	3.8	2.7	0.089
Stroke severity score	6.8	6.8	0.934	6.7	7.1	0.621
Thrombolytic therapy	2.6	2.2	0.367	3.5	1.7	0.003
Endovascular therapy	1.4	1.8	0.346	1.3	1.3	1.000
Reperfusion therapy	4.0	4.0	1.000	4.8	3.0	0.068
In-hospital mortality	0.6	0.9	0.247	0.9	1.2	0.464
Length stay (days)	10.8	8.4	0.001	9.8	8.0	0.009

EMS: Emergency medical services; Jan: January; Confi2019: period January to May 2019; Confi2020: period January to May 2019 (corresponding to state of alarm).

**Table 3 brainsci-11-00092-t003:** Correlations matrix of main variables of study and number of affected by COVID-19 in Extremadura, Spain (COVID-19 period).

Variable	COVID	Code	All-S	IS	HS	TIA	T-S	TT	EVT	RT
COVID	.	−0.090	0.198	0.060	0.349	−0.026	0.165	−0.409 *	−0.455 *	−0.525 *
Code	−0.090	.	0.392	0.391	−0.080	0.230	−0.099	0.422	−0.027	0.219
All-S	0.198	0.392	.	0.810 *	0.267	0.468 *	0.452 *	0.203	0.021	0.127
IS	0.060	0.391	0.810 *	.	−0.237	0.082	0.569 *	0.285	0.122	0.239
HS	0.349	−0.080	0.267	−0.237	.	0.102	0.035	−0.184	−0.231	−0.253
TIAs	−0.026	0.230	0.468 *	0.082	0.102	.	−0.192	0.089	0.041	0.077
T-S	0.165	−0.099	0.452 *	0.569 *	0.035	−0.192	.	0.283	0.382	0.407
TT	−0.409 *	0.422	0.203	0.285	−0.184	0.089	0.283	.	0.362	0.796 *
EVT	−0.455 *	−0.027	0.021	0.122	−0.231	0.041	0.382	0.362	.	0.853 *
RT	−0.525 *	0.219	0.127	0.239	−0.253	0.077	0.407	0.796 *	0.853 *	.

COVID: number of affected by COVID−19, Code: Stroke code activations, All-S: All stroke admissions, IS: Number Ischemic stroke, HS: Number Hemorrhagic stroke, TIA: Number Transient Ischemic Attack, T-S: Number Tele-stroke consultations, TT: N° Thrombolytic therapy, EVT: N° Endovascular therapy, RT: N° Reperfusion therapy. *p* < 0.05 with *.

## Data Availability

All the data with which this work has been elaborated are available for any researcher under reasonable and comprehensible request to the corresponding author (mailto: josemaria.ramirez@salud-juntaex.es).

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
