# Peer review of "Slump in Hospital Admissions for Stroke, a Fact of an Uncertain Nature That Requires Explanation"

_brainsci, 2021, doi:10.3390/brainsci11010092_

Round 1

Reviewer 1 Report

The study by Ramírez-Moreno et al. is very interesting. Undoubtedly, the  COVID-19  pandemic is a great challenge for all of us, however it has the greatest impact  on  the healthcare system, which as the Authors stated “in many cases leading to a situation of collapse”.

Some issues should be raised before publishing:

  1. Methods section should be divided into subsections for greater clarity.
  2. A similar remark applies to the Results section. Structuring the results will allow readers to have a better perception.
  3. Be consistent and use the same abbreviations throughout your text, tables, and figures, e.g you used confi2020 in the text while conf2020 in the table.
  4. Figures require resolution improvement.They are unreadable in the present form.
  5. Page 3, the Authors stated: “The study protocol was submitted to the local Ethics Committee and meets the state legal requirements in the field of biomedical research, personal data protection and good clinical practice standards, also as stated in the Declaration of Helsinki.” whereas on page 9 the Authors stated: “The study  protocol  was not submitted  to  the  local  Ethics Committee and meets the state legal requirements in the field of biomedical research, personal data protection and good clinical practice standards, also as stated in the Declaration of Helsinki.”. Please leave the correct information.
  6. Check the format for references according to Instructions for authors.

Author Response

Reviewer: 1

We appreciate the Reviewer’s input to review the first version and give this positive comment.

Comments to the Author

  1. Methods section should be divided into subsections for greater clarity.

We welcome the reviewer's suggestion and agree that it would be useful to structure the text to improve the readability of the document. We decided to make some changes to the manuscript. We hope it was the right decision.

  1. A similar remark applies to the Results section. Structuring the results will allow readers to have a better perception.

In the same vein as the previous section, we decided to make some changes to the manuscript to better structure the results section.

  1. Be consistent and use the same abbreviations throughout your text, tables, and figures, e.g you used confi2020 in the text while conf2020 in the table.

We appreciate the consultation. Now we have corrected this aspect. We review the abbreviations throughout the text, tables and figures.

  1. Figures require resolution improvement. They are unreadable in the present form.

We apologize for the lack of information in the Figures and have improved the resolution. We believe that now, they are legible in their current form.

  1. Page 3, the Authors stated: “The study protocol was submitted to the local Ethics Committee and meets the state legal requirements in the field of biomedical research, personal data protection and good clinical practice standards, also as stated in the Declaration of Helsinki.” whereas on page 9 the Authors stated: “The study  protocol  was not submitted  to  the  local  Ethics Committee and meets the state legal requirements in the field of biomedical research, personal data protection and good clinical practice standards, also as stated in the Declaration of Helsinki.”. Please leave the correct information.

We appreciate this opportunity to clarify this point. The correct information is:

 “The study  protocol  was not submitted  to  the  local  Ethics Committee and meets the state legal requirements in the field of biomedical research, personal data protection and good clinical practice standards, also as stated in the Declaration of Helsinki.”.

  1. Check the format for references according to Instructions for authors.

Thank you for this suggestion. We review the references according to the journal's standards (American Chemical Society (ACS) Citation Style). References are numbered in order of appearance in the text and individually numbered at the end of the manuscript.

Reviewer 2 Report

Authors reported the decline in stroke admissions during the first five months of 2020 compared with the corresponding period of 2019 in a Spanish region. Authors found that stroke activation codes increased, while the number of stroke admissions significantly decreased.

Some points of the paper can be improved. Please find my observations below.

  • Stroke admissions decreased in women more than in men. Do Authors have any explanations for that?
  • Please note that there are several available reports on changes in stroke admissions between 2019 and 2020 (see for example Stroke 2020;51:3746-3750; Clin Neurol Neurosurg 2020;201:106436). It would be useful to discuss the difference between the Authors’ findings and those of other studies.
  • From Figure 1-B, the reader can see that stroke admissions decreased from March 2020 onwards, while staying higher than in the corresponding period of 2019 until March. This is in my opinion an interesting finding, as it suggests a high impact on lockdown, rather than stroke incidence itself, in the decrease in hospital admissions. This point needs further discussion.
  • A limitation of the present study – as well as of most papers in the field – was that it did not consider patients treated at home. Several stroke cases might have been managed at home during the pandemic. This is a point to discuss.
  • It would be useful to show a comparison between stroke admissions in larger vs smaller hospitals. Several small hospitals might have been converted to “Covid hospitals” during the pandemic, therefore limiting the admissions of stroke patients.
  • Please explain the legend in Figure 2. For example, I cannot understand the meaning of “TT”.

Author Response

Reviewer: 2

We appreciate the Reviewer’s input to review the first version and give this positive comment

  1. Stroke admissions decreased in women more than in men. Do Authors have any explanations for that?

The issue of gender differences in stroke is not new and this findings are not surprising. It is possible that socioeconomic factors...including marital status played a role in the outcomes of this study. Because women live longer than men, they may live alone at the time a stroke occurs and may be socially isolated.

Better understanding of the type and degree of inequities in stroke management and outcomes are important for improving health outcomes, and an initial step towards personalized medicine.

  1. Please note that there are several available reports on changes in stroke admissions between 2019 and 2020 (see for example Stroke 2020;51:3746-3750; Clin Neurol Neurosurg 2020;201:106436). It would be useful to discuss the difference between the Authors’ findings and those of other studies.

Thank you for this suggestion. We agree on the needs to include references about the difference between our findings and those of other studies.  We have included the following references:

- Sacco S, Ricci S, Ornello R, Eusebi P, Petraglia L, Toni D; Italian Stroke Organization. Reduced Admissions for Cerebrovascular Events During COVID-19 Outbreak in Italy. Stroke. 2020 Dec;51(12):3746-3750. doi: 10.1161/STROKEAHA.120.031293

- Siegler, J. E.; Zha, A. M.; Czap, A. L.; Ortega-Gutierrez, S.; Farooqui, M.; Liebeskind, D. S.; Desai, S. M.; Hassan, A. E.; Starosciak, A. K.; Linfante, I.; Rai, V.; Thon, J. M.; Then, R.; Heslin, M. E.; Thau, L.; Khandelwal, P.; Mohammaden, M. H.; Haussen, D. C.; Nogueira, R. G.; Jillella, D. V; Nahab, F.; Kaliaev, A.; Nguyen, T. N.; Zaidat, O.; Jovin, T. G.; Jhadav, A. P. Influence of the COVID-19 Pandemic on Treatment Times for Acute Ischemic Stroke: The Society of Vascular and Interventional Neurology Multicenter Collaboration. Stroke 2020. https://doi.org/10.1161/STROKEAHA.120.032789.

  1. From Figure 1-B, the reader can see that stroke admissions decreased from March 2020 onwards, while staying higher than in the corresponding period of 2019 until March. This is in my opinion an interesting finding, as it suggests a high impact on lockdown, rather than stroke incidence itself, in the decrease in hospital admissions. This point needs further discussion.

We appreciate the reviewer's suggestive suggestion and agree that further discussion would be helpful.

Exactly in Figures 1, the reader can see that the activation of the stroke code increased in 2020 compared to the previous year by 19% (Figure 1A), but this number of activations during the 10 weeks of strict confinement decreased significantly. very significant, and there was not the expected increase in activations for this period compared to the same period of the previous year (graph 1B).

The cause of the observed decrease in stroke activations is unknown. This observation could be attributed to a true decrease in the incidence of strokes or to patients who did not seek emergency medical care during the pandemic. Our study supports some of the earlier observations of the SARS outbreak in 2003, which resulted in a marked reduction in the number of emergency department visits. The fear of contracting COVID-19 from going to hospitals may have led to a reduction in stroke presentations. In this sense, our observation that patients with milder strokes would stay at home would also go, we see a change in the severity of stroke presentations. These aspects are explained in the discussion.

  1. A limitation of the present study – as well as of most papers in the field – was that it did not consider patients treated at home. Several stroke cases might have been managed at home during the pandemic. This is a point to discuss.

Yes, we fully agree with the observation. One limitation of our study is that we do not have information on those patients who could have been cared for at home or died at home due to a stroke without being cared for.

Yes, several cases of stroke may have been treated at home during the pandemic, but we do not have that data and our records are not aligned with top-tier health care records. We have added this potential limitation to our work.

  1. It would be useful to show a comparison between stroke admissions in larger vs smaller hospitals. Several small hospitals might have been converted to “Covid hospitals” during the pandemic, therefore limiting the admissions of stroke patients.

That analysis would be interesting, but it is not exactly what happened in our region, nor does our data allow that specific analysis. Hospitals in the region have implemented contingency plans to contain the pandemic, which include the suspension of non-essential visits and the adjustment of clinical services for outpatients and inpatients, as well as the progressive opening of quarantine rooms according to needs. of attention. Within this overall planning, stroke units in the region were not limited in capacity, and the inpatient stroke unit staff remained unchanged during the height of the pandemic. Multimodal brain imaging, CT angiography and endovascular therapy were ensured to be available 24 hours a day, 7 days a week for eligible patients. There were also no limitations on the use of telestroke, and all 9 hubs were operational. There were no relevant changes in the stroke care protocol.

  1. Please explain the legend in Figure 2. For example, I cannot understand the meaning of “TT”.

We apologise for the lack of information in the Figure 2. The note to Figure 2 includes now the explanation of TT, TR and EVT.